# A Network Model Approach to International Aid

**DOI:** 10.3390/e25040641

**Published:** 2023-04-11

**Authors:** Joe Scattergood, Steven Bishop

**Affiliations:** Department of Mathematics, University College London, London WC1E 6BT, UK

**Keywords:** international aid, foreign aid, complex systems, network science, network theory, econometrics, financial mathematics, portfolio theory

## Abstract

Decisions made by international aid donors regarding the allocation of their aid budgets to recipients can be mathematically modelled using network theory. The many countries and multilateral organisations providing developmental aid, mostly to developing countries, have numerous competing or conflicting interests, biases and motivations, often obscured by a lack of transparency and confused messaging. Using network theory, combined with other mathematical methods, these inter-connecting and inter-dependent variables are identified, revealing the complicated properties and dynamics of the international aid system. Statistical techniques are applied to the vast amount of available, open data to first understand the complexities and then identify the key variables, focusing principally on bilateral aid flows. These results are used to create a weighted network model which is subsequently adapted for use by a hypothetical aid recipient. By incorporating modern portfolio theory into this weighted network model and taking advantage of a donor’s reasons for allocating their aid budgets to that recipient, a simulation is carried out treating the problem as an optimal investment portfolio of aid determinant ‘assets’ which illustrates how a recipient can maximise their aid receipts. Suggestions are also made for further uses and adaptations of this weighted network model.

## 1. Introduction

US$162bn of foreign aid was donated by developed countries (‘donors’) to developing countries (‘recipients’) in 2020 [1]. Democratic governments of donor countries are faced with decisions regarding how and where to allocate their foreign aid budgets, not solely for poverty alleviation but also to achieve a diverse set of specific goals and unique strategies.

Significant drivers of how donors allocate their foreign aid budgets are based upon achieving certain political and strategic objectives, both domestic and global. Global objectives include the projection of soft power, control over foreign resources, biases towards allies or ex-colonies and gaining global influence. These motivations, behaviours and determinants are complicated and difficult to capture in a mathematical model. Nevertheless, in democratic societies at least, justifications for aid allocation decisions and transparency are often demanded, and mathematical methods and models can help provide these, even if they are not used to determine forward action.

Foreign aid dynamics and interactions are particularly complicated, as detailed in Ben Ramalingam’s book, *Aid on the Edge of Chaos* [2], with the complexity of the determinants of aid flows well documented by [3]. There are many interacting variables and dynamics of foreign aid networks, and the relative importance which donors place on their specific and numerous aid determinants is often not known. This causes difficulties when analysing and concluding on many aspects of overseas aid, which further makes it problematic to design and create a useful mathematical model that can capture the dynamics of foreign aid networks and successfully incorporate the interacting and inter-dependent variables.

To attempt this, the many variables and determinants which create the complicated foreign aid dynamics firstly need to be identified and understood. By studying other research conducted on this topic, and sourcing and analysing additional data using statistical techniques, the results from this article will inform the adaptation and use of a weighted network model, first proposed by [4], that captures the properties, interactions and dynamics of the international aid system.

Much of the literature and research into foreign aid dynamics focuses on the donor. Econometric techniques, primarily regression and ordinary least squares ([5,6,7]), are commonly employed in an attempt to reveal the relative importance of donor motivations and potential biases behind their aid allocation decisions. More recently, there has been research conducted on the growing field of network theory and the utilisation of related mathematical methods to model the allocation of aid that goes beyond regression ([4,8]).

However, it is rarer to find research and analysis focusing on aid recipients. By understanding donor motivations and biases, aid recipients could exploit these ‘assets’ and potentially increase their aid receipts if they are viewed as a portfolio of investments.

By identifying and quantifying significant donor motivations for allocating their aid budgets, inputting these variables into a general weighted network model [4] and then adapting it using financial mathematics (modern portfolio theory), aid recipients could use the model to optimise their aid income portfolio, treating donor variables similarly to assets in an investment portfolio. This is illustrated in this article using a simulation.

The principal aim, then, of this article is to illustrate the power of network science and mathematical modelling when applied to the complex and dynamical system of international aid. The potential impact is an increase in transparency of the often-opaque motivations and biases of aid donors, which subsequently could be employed by recipients to increase their aid income.

## 2. Methods

### 2.1. Data and Data Analysis

The first step to evaluating, and then adapting, the general weighted network model [4] is to identify the significant motivations and preferences shown by selected donors regarding the allocation of their aid budgets. These will be used as the model’s variables. Subsequently, the accuracy of the model’s mechanics and outputs can be tested against actual historical data for selected donors and recipients.

Data from the OECD and World Bank were sourced and analysed using various statistical techniques to identify and understand the inter-connecting and inter-dependent variables that drive the data. The pertinent results are summarised here.

#### 2.1.1. Economic and Foreign Aid Data

To compare the economic fortunes of one country versus another, gross national income (GNI), a key measure of economic well-being and a superior metric for assessing the overall economic condition of a country, especially for countries that have large foreign receivables or outlays, will be used for identifying the level of need of an aid recipient (‘recipient need’). Furthermore, to assist with country comparisons, GNI per capita will be used rather than absolute GNI. 

Table 1 lists the top 10 aid recipients in 2019 by net official development assistance (ODA) receipts, classified as total net ODA flows from Development Assistance Committee (DAC) countries, multilateral organisations and non-DAC countries.

When identifying the top donor countries, rather than looking at absolute aid donated, the affordability of a donor country to provide aid is assessed using aid donated as a percentage of country GNI. This is summarised in Figure 1, which lists the members of the DAC, a development committee of the OECD.

Figure 1 is sourced from the OECD website [10] and arranged in descending order based on the percentage of DAC-country GNI donated in 2020, with Sweden donating the highest percentage of its GNI at 1.15%.

Donor affordability is epitomised by the 0.7% target agreed by the United Nations (UN) in 1970 for aid contributions by DAC countries to developing countries. It is reasonable then to assume that since the UN members agreed to 0.7%, they can therefore afford to donate 0.7% of their GNI. However, as shown in Figure 1, this target is not being met by most UN countries, including the USA.

Bilateral aid flows—aid given directly from a country donor to a recipient donor—comprised circa. 67% of total ODA donated in 2019, with the remaining third being flows from multilateral institutions and international financial institutions (for example, the World Bank). However, the proportion of bilateral aid reduced significantly in 2020 by 36% on 2019 levels to 42% of the total aid donated, with multilateral institutions taking up the slack, due mainly to the impact of COVID [11].

#### 2.1.2. Aid Flows from Donors to Former Colonies

There are robust conclusions in the research performed by [3,5,6], among many others, that a strong motivation behind aid allocation decisions by donors lies in whether the recipient is an ex-colony or not. Former colonies receive proportionately more aid from their former colonial masters than other recipients.

Indeed, according to [5], between 1970 and 1994, France gave 57% of its total bilateral aid to its former colonies, the UK gave 78% and Portugal 99.6%. Moreover, according to the OECD [12], in 2009, the largest recipient of UK aid was India and, by 2019, this was Pakistan, both former UK colonies. Thus, colonial history is positively correlated with aid, as identified by [5] and confirmed by own analysis performed.

#### 2.1.3. Trade Activity

Before correlation techniques were applied to detect any interdependencies between trade activity and aid donations, the raw data were analysed. Trade data are sourced from the World Integrated Trade Solution (WITS) website, a sister site of the World Bank specifically focused on trade [13], for the period 1993 to 2019. By charting this trade data with aid data sourced from the World Bank [9], a pattern of aid versus trade can be viewed over time. This suggested a positive correlation, confirmed by calculating correlations between the two data sets over many periods. This result is also backed by research performed by [5,6,14].

#### 2.1.4. Recipient Need

The literature is mixed regarding the relative importance of recipient need as a variable in a donor’s aid allocation decisions. In [5], the authors are clear on donor motivations being based mainly on self-interest and political and strategic considerations over aid recipient needs. However, later studies, such as [14], dispute this conclusion stating that self-interest, while still a significant input into aid allocation decisions, is not as important as recipient need. Moreover, [6] conclude that the USA behaves very differently from all other aid donors, except Japan, by putting much less emphasis on recipient need and much more emphasis on donor self-interest.

#### 2.1.5. The Herding Phenomenon (the Bandwagon Effect)

Another variable to consider for inclusion in a mathematical model of foreign aid is herding behaviour often exhibited by donors, also termed the ‘bandwagon effect’. This refers to the actions and impulses of a group of agents, countries, politicians, or financial traders to follow the actions of the ‘crowd’ rather than trust their own individual judgment. The phenomenon has similar attributes to ‘groupthink’. It is an emergent behaviour of a dynamical system due to the many interactions taking place within that system.

Herding is commonly associated with financial market behaviour, for example asset bubbles [15]. Grounded in behavioural finance, herd mentality refers to investors’ bias to follow what other investors are doing, being largely influenced by emotion and intuition, rather than by their own evaluations of potential investments.

In terms of aid allocation, the bandwagon effect manifests itself when a recipient receives more aid from one donor, leading to an increase in aid from many more donors. In other words, the more aid a recipient receives, the more it attracts. It likely depends on the relative influence of the lead donor rather than characteristics of the recipient.

Research conducted by [6] attempted to measure the effect using regression and incorporating aid from other sources, not only ODA. They find that there is some support for the herding argument, but it is far from conclusive. Ref. [16] gives the phenomenon a more thorough review, concluding that there is around an 11% impact on aid donations through donor herding, which is relatively significant.

### 2.2. A Network Model for Foreign Aid

The principal outcome of the research conducted, using data analysis and statistical techniques, is the identification of the following significant aid determinants:Past colonial relationships;Trade activity and commercial interests;Poverty alleviation (recipient need); andBandwagon impacts (‘herding’).

These variables will now be incorporated into a weighted network model to demonstrate how such a model can be modified and used by an aid recipient to treat their various donor-sourced aid receipts as an investment portfolio and maximise their aid income using modern portfolio theory.

The weighted network model introduced allows for additional variables to be incorporated and, indeed, further variables were considered for inclusion. Variables such as the occurrence of war, migration and recipient corruption could be reflected in the model; however, the focus here is on long-term and relatively stable determinants of aid. Furthermore, the bandwagon impact may partially and indirectly incorporate these variables; for example, the war in Afghanistan in the early 2000s led to significant amounts of aid donated by the USA to Afghanistan, swiftly followed by aid donated by other donors.

#### 2.2.1. The General Weighted Network Model

The general model proposed by [4] follows a weighted network model approach utilising donor-specific preference functions to measure donor motivations and biases when deciding aid allocations. The preference functions quantify the *relative* contributions of aid determinants used in aid allocation input decisions, such as poverty, trade activity, past colonial relationships and bandwagon impacts (‘herding’), into ‘weights’ which are applied to a network model, revealing donor behaviours and the relative importance placed on these aid determinants.

Figure 2 is an archetypal bipartite network model which has nation donors on the left representing the set of nodes D and the recipients on the right representing the set of nodes R. D and R are disjoint sets of nodes in which links can exist only between the two sets and not within each set, thus illustrating the flow of aid which is directed only from elements of D to elements of R. In total, there are six nodes split into two disjoint sets of three donors, $d_{i}\in D$, and three recipients, $r_{j}\in R$, where i,j=1,2,3 represent donors and recipients, respectively.

The n-vector node-specific information $u_{\beta }^{k}$ represents the n quantities (or vector elements) associated with each country, k=i,j, in the network, where β=1,…,n is used to denote the numbered element of the vector $u^{k}$. For example, recipient Ethiopia’s node in Figure 2 is labelled $r_{3}$. In the case that this node’s specific information contains poverty levels, $u_{1}^{3}$, colonial history, $u_{2}^{3}$, and trade activity, $u_{3}^{3}$, then the vector $u^{3}$ has 3 elements n=3, denoted as $u_{\beta }^{3}$, where β=1,2,3.

Further, the links between each set holds m-vector link weights $l_{\alpha }^{ij},$ representing the m relationships between donors i and recipients j, where α=1,…,m denotes the numbered element of the vector $l^{ij}$. For example, the vector representing trade activity and colonial history between the UK, $d_{2}$, and Bangladesh, $r_{1},$ are denoted as $l_{1}^{21}\;\mathrm{and}\;l_{2}^{21}$, respectively, where m=2 in this example, and is denoted as $l_{\alpha }^{21}$, where α=1,2.

The node and link vector information, as defined, can be quantified into weights and input into a preference Function (1), supplemented by input Functions (2) and (3), outputting a percentage of aid allocated by a donor, $d_{i}$, to a recipient, $r_{j}$.
(1)$$P^{ij}\left(l^{ij},u^{j}\right):=\prod_{\alpha }f_{\alpha }^{i}\left(l_{\alpha }^{ij},l_{\alpha }^{i•}\right).\prod_{\beta }g_{\beta }^{i}\left(u_{\beta }^{j},u_{\beta }^{•}\right)$$
The superscript • in preference Function (1) denotes all recipients in the set of nodes R, and its role is seen in the denominators of (2) and (3). The input functions, $f_{\alpha }^{i}$ and $g_{\beta }^{i}$, quantify donor preferences towards specific determinants of aid, such as trade activity and recipient need, into proportioned weights before input into (1):(2)$$f_{\alpha }^{i}\left(l_{\alpha }^{ij},l_{\alpha }^{i•}\right)={\left[\frac{l_{\alpha }^{ij}}{\sum_{k\in V_{2}}l_{\alpha }^{ik}}\right]}^{\mu_{\alpha }^{i}}$$
(3)$$g_{\beta }^{i}\left(u_{\beta }^{j},u_{\beta }^{•}\right)={\left[\frac{u_{\beta }^{j}}{\sum_{k\in V_{2}}u_{\beta }^{k}}\right]}^{\eta_{\beta }^{i}}$$
where the exponent parameters, $\mu_{\alpha }^{i}$ and $\eta_{\beta }^{i}$, hold only non-negative real values. These are referred to as ‘power’ parameters.

The terms in the square brackets in (2) and (3) are functional inputs holding information on chosen aid determinants positively correlated with aid allocation and expressed as a proportion. The greater the proportion, the higher the value generated by the function and therefore the greater the weight due to a particular aid determinant for input into (1).

There is a difference in usage between the input Functions (2) and (3). The function $f_{\alpha }^{i}\left(l_{\alpha }^{ij},l_{\alpha }^{i•}\right)$ in (2) is used for link-specific weights, $l_{\alpha }^{ij}$, and quantifies behaviours and relationships that exist between a donor $d_{i}$ and a recipient $r_{j},$ for example the levels of trade activity. The function $g_{\beta }^{i}\left(u_{\beta }^{j},u_{\beta }^{•}\right)$ in (3) is used for node-specific weights, $u_{\beta }^{j}$, and quantifies a specific recipient metric, such as the poverty ratio among recipients, which is a determinant of aid that bears no direct relationship to a particular donor. Note $u_{\beta }^{j}$ in (3) is specific to a recipient $r_{j}$; however, it can also be specific to a donor, $u_{\beta }^{i}$, to quantify determinants specific to a donor, $d_{i}$, such as donor affordability.

The form of (2) and (3) assumes that a positive correlation exists between the determinant in question and aid allocated. For negative correlations, these terms are modified by subtracting the functions from (1), resulting in a recipient with a lesser proportion receiving a higher weight of preference and therefore aid allocated relative to the other recipients in the model.

As an example, assume ‘recipient need’ was chosen to be an aid determinant by a particular donor. This variable can be measured in several ways. First, say it is measured by poverty levels per capita. This measure is assumed positively correlated with aid: the more people in poverty, then the more aid the recipient should attract, resulting in a relatively higher proportional output by Equation (3)—the recipient-specific weight—assuming the power parameter $\eta_{\beta }^{i}$ is unity. The output from (3) is then an input into the preference Function (1), resulting in a higher percentage of aid to that recipient. On the other hand, if recipient need was instead measured using recipient GNI, the output of Function (3) would need to be subtracted from (1) because recipient GNI is assumed negatively correlated with aid received. The result of these two approaches in terms of the output by preference Function (1) should be roughly equal.

To allow for biases when allocating aid to certain recipients based on aid determinants, the power parameters $\mu_{\alpha }^{i}$ and $\eta_{\beta }^{i}$ on Functions (2) and (3), respectively, allow for a choice to be made by a donor with regards to the relative contribution and, thus, importance of particular determinants on the final aid allocations output by (1), which is in the form of percentages of the total aid budget. Donors can dial up or dial down the level of influence that their selected determinants have on the outcome by changing the values of the power parameters.

For example, if a donor wanted to allocate more aid to recipients with which it experiences large amounts of trade activity over those recipients with higher poverty levels, then the donor will choose a higher value for the power parameter applicable to the relevant functional equation that is quantifying trade activity. These parameters then, also provide a means for deducing historical donor behaviours and biases in simulations.

If a mathematical model is to be used by politicians, countries and organisations, then it needs to be simple, effective and able to be communicated. An important feature of this weighted network model is indeed its simplicity and transparency with the inputs into Equations (2) and (3) and, in turn, into preference Function (1), determined by verifiable, properly sourced, factual data.

The weighted network model’s initial purpose was to reflect the decisions made by donors with regards to their motivations towards aid allocation based on certain factors, such as trade and recipient poverty. Donors can decide how much emphasis these factors have on the final allocation of their aid budgets. With historical data, sourced from the World Bank and OECD, input into Functions (2) and (3), and with the historical aid allocation figures which are the outputs from preference Function (1) also known, this leaves the power parameter values as the only unknowns. These values can be determined by playing the role of balancing figures and, from these estimated values, donor motivations and the relative importance placed on certain aid determinants can be studied.

#### 2.2.2. Adaptation of the General Model

Before adaptation of the general weighted network model for use by an aid recipient, the model Functions (1)–(3) need to first reflect the analysis conducted in Section 2.1 and the network model in Figure 2. Therefore, three donors, three recipients and the four identified significant aid determinants are to be incorporated into the general model.

The set of donors, D, are Germany, the UK and the USA, respectively; $d_{1},d_{2},d_{3}\in D$, each having its own preference function. The set, R, contains the three recipients: Bangladesh, Afghanistan and Ethiopia, respectively; $r_{1},r_{2},r_{3}\in R$.

From analysis performed, trade relationships were found to be positively correlated with aid and bilateral trade activity was identified as one of the four significant aid determinants that a donor considers when allocating their aid budgets. For incorporation into the general model, trade activity is to be represented by the variable $t^{ij}.$ For example, the superscript i=1 identifies the donor as Germany, $d_{1},$ and superscript j=1 represents the recipient Bangladesh, $r_{1}$. The trade ratio between a donor and its aid recipients, measured in terms of exports to the recipient in US$, was calculated from [13]. For example, for Germany, $t^{11}:t^{12}:t^{13}$ ≡ 13:1:5 for the year 2019.

Recipient poverty was another significant aid determinant identified and is to be represented in the adapted model by $p^{j}$, which denotes the poverty levels in recipient $r_{j}$. The ratio of poverty levels is determined by using GNI per capita to quantify recipient need [9]. Correlation analysis indicated that this metric is negatively correlated with aid; therefore, there will be a subtraction from 1 in the poverty-specific functional input equation.

Another of the significant aid determinants identified was colonial relationships, represented by the variable $c^{ij}$ between donor $d_{i}$ and recipient $r_{j}$. This variable will be quantified using a binary zero-one integer programming variable defined as
(4)$$c^{ij}=1+\left\{\begin{array}{l} 1,\;\mathrm{colonial}\;\mathrm{relationship}\;\mathrm{existed}\;\mathrm{between}\;d_{i}\;\mathrm{and}\;r_{j}\\ 0,\;\mathrm{no}\;\mathrm{colonial}\;\mathrm{relationship} \end{array}\right.$$
The only colonial relationships of relevance to the simulation are Afghanistan and Bangladesh, both ex-UK colonies and protectorates, and thus $c^{21}=2$ and $c^{22}=2$; with $c^{ij}=1$ for all other combinations of i and j, where i,j=1,2,3.

The final significant aid determinant identified was the bandwagon effect, or herding, discussed in Section 2.1.5. Simply, it refers to the tendency of aid donors to follow other donors in allocating aid to certain recipients, who then gain ‘star’ status in the network. This can reveal itself when a recipient attracts a larger proportion of the total aid donated for no discernible reason, controlling for other factors; see [16]. The weighted network model framework can quantify the bandwagon effect, to be denoted $b^{j}$, by capturing the phenomenon using aid received by a recipient, $r_{j},$ in the previous period as a proportion of total aid donated by all donors in the entire network. This captures the herding effect by measuring recipients’ previous success in receiving aid relative to other recipients, thus becoming a ‘star’ node in the model network.

The four determinants have now been allocated specific variables and associated data to be input into an adapted model. The aid allocated by the three donors to the three recipients in Figure 2 is to be used as the output values of the adapted model’s preference function, sourced from [9]. Thus, the only remaining unknowns are the values of the four power parameters in the four input equations, each representing one of the four aid determinants. These parameter values can be estimated by running the model using the known inputs (the aid determinants) and known outputs (actual historical data) to provide important insights into donors’ individual and relative motivations and behaviours with regards to allocating their aid budgets. The higher the power parameter value, the more bias has been baked into the aid allocation output from that aid determinant.

To use the model over multiple consecutive time periods, it needs to be made temporal. Starting at time t=1, the weighted network model can be iterated forward in time with the outputs of the equations changing at each t due to the recipients’ economic response to aid receipts, which feed back into donors’ decisions on the allocation of aid at t+1, acting as a feedback mechanism. For example, assume aid donated at time t led to a reduction in poverty in a recipient. By rolling the model forward to the next time-period, t+1, this reduced level of poverty will be fed into the model at t+1, producing a different aid allocation output percentage for the donor at t+1 compared to t.

By denoting recipient poverty as $p_{t}^{j}$, trade relationships as $t_{t}^{ij}$, and the bandwagon effect as $b_{t}^{j}$ where the subscript t represents the time-period, the model becomes dynamic with respect to time. Note that colonial history, $c^{ij}$, quantified in (4) is a static measure: it does not change with time and, consequently, has no subscript.

After the alterations discussed, the preference Function (1) and input Functions (2) and (3) in the general model have been adapted to create Equations (5)–(9), with the four determinant functions in the preference Function (5) and four input Equations (6)–(9).
(5)$$P^{ij}\left(c^{ij},t_{t}^{ij},p_{t}^{j},b_{t}^{j}\right):=f_{1}^{i}\left(c^{ij},c^{i•}\right)\cdot f_{2}^{i}\left(t_{t}^{ij},t_{t}^{i•}\right)\cdot g_{1}^{i}\left(p_{t}^{j},p_{t}^{•}\right)\cdot g_{2}^{i}\left(b_{t}^{j},b_{t}^{•}\right)$$
(6)$$f_{1}^{i}\left(c^{ij},c^{i•}\right)={\left[\frac{c^{ij}}{\sum_{k\in V_{2}}c^{ik}}\right]}^{\mu_{1}^{i}}$$
(7)$$f_{2}^{i}\left(t_{t}^{ij},t_{t}^{i•}\right)={\left[\frac{t_{t}^{ij}}{\sum_{k\in V_{2}}t_{t}^{ik}}\right]}^{\mu_{2}^{i}}$$
(8)$$g_{1}^{i}\left(p_{t}^{j},p_{t}^{•}\right)={\left[1-\frac{p_{t}^{j}}{\sum_{k\in V_{2}}p_{t}^{k}}\right]}^{\eta_{1}^{i}}$$
(9)$$g_{2}^{i}\left(b_{t}^{j},b_{t}^{•}\right)={\left[\frac{b_{t}^{j}}{\sum_{k\in V_{2}}b_{t}^{k}}\right]}^{\eta_{2}^{i}}={\left[\frac{\sum_{m=1}^{3}A_{t-1}^{mj}}{\sum_{k=1}^{3}\sum_{n=1}^{3}A_{t-1}^{nk}}\right]}^{\eta_{2}^{i}}$$
where $A_{t-1}^{mj}$ in (9) is the amount of aid donated by donor m to recipient j in the previous period t−1, and the denominator of (9) quantifies the total aid donated within the network at period t−1.

The model is iterated forward starting from t=1 (year 2015) to t=5 (year 2019). Note that t−1 is the year 2014, for which actual aid data is input into Function (9). By iterating forward, the power parameters can be backward calculated for each year from 2015 to 2019. The values are shown in Table 2. Entries marked N/A for not applicable are included where the relationship was not relevant for the year in question.

The values in Table 2 can be put into matrix form. For each year the model was iterated, the matrix of power parameter values was input into (6)–(9) and fed into (5) to create the next period’s aid allocations. For example, for the year 2015, the matrix was the following:(10)$$\Phi_{\alpha }^{i}=\left(\begin{array}{cccc} \mu_{1}^{1} & \mu_{2}^{1} & \eta_{1}^{1} & \eta_{2}^{1}\\ \mu_{1}^{2} & \mu_{2}^{2} & \eta_{1}^{2} & \eta_{2}^{2}\\ \mu_{1}^{3} & \mu_{2}^{3} & \eta_{1}^{3} & \eta_{2}^{3} \end{array}\right)=\left(\begin{array}{cccc} 0.0 & 1.5 & 1.1 & 4.0\\ 0.0 & 0.4 & 0.0 & 1.0\\ 0.0 & 0.0 & 0.4 & 1.0 \end{array}\right)$$

The general weighted network model represented by Equations (1) to (3) has been tested, modified to Equations (5) to (9), and the relevant power parameters now calculated. All significant inputs and outputs for the years 2015 to 2019 are now known. Next, the model is adapted for use by an aid recipient by incorporating modern portfolio theory before performing a simulation to illustrate how that recipient could optimise their aid receipts.

#### 2.2.3. An Aid Recipient’s Investment Strategy Using Network Theory

If a recipient invests in increasing its trade activity with donors, then that recipient should expect an increase in aid receipts from those donors who place a relatively high value on commercial trade in their aid allocation decisions. This increase would then be compounded by the herding effect, leading to additional receipts that can be re-invested back into trade activity or other similar investment ‘assets’, creating a virtuous cycle of investment and increasing returns.

Paradoxically, recipients may not have an incentive to reduce poverty since it may lead to a fall in aid receipts. Instead, if recipients focus primarily on increasing trade activities, then their GNI should naturally increase and poverty should be reduced. This argument is limited however as it depends on other limiting factors such as the quality of governance and institutions in the recipient country. The fruits of increased trade activity may also fuel corruption rather than being devoted to alleviating poverty. Often, increased trade activity is performed by state-owned companies with the recipient’s President as the main shareholder. Donors may wish to accommodate this in their aid decisions, which the weighted network model can do.

Despite these complications, the main interest here is regarding the ability of the weighted network model, illustrated by Figure 2 and Equations (5)–(9), to be used by a recipient as an investment tool to maximise their aid receipts.

By creating a foreign aid network model, a recipient would initially discover how influential it is in the network using centrality measures, the links it holds with donors and those that it does not. Specific weights can be added to links and nodes containing proportions of aid received, trade activity and other recipient–donor dyad information. This network model could also indicate if the recipient should seek out new donors, invest in current donors or a combination of the two.

Recipients can treat their aid network model much like a company seeking to attract funding. They could view the aid determinants used by donors as an ‘asset portfolio’, safeguarding and maximising the value of those assets by treating them as investments. Recipients can invest their aid income into the asset portfolio, for example by investing in trade relationships with donors. The recipient may also need to invest in other sub-activities such as governance quality and public relations activities, which the network model and portfolio can identify.

An investment plan for a typical aid recipient is outlined as follows:

Step (1): Create a weighted network model, providing insights into links, level of influence and current donors in the recipient’s foreign aid network. Analyse each donor’s aid determinant preferences, motivations and biases.

Step (2): Produce an asset portfolio representing the donor preferences identified, e.g., trade activity and poverty alleviation, with the USA being highlighted as a highly influential donor.

Step (3): Identify those assets in the portfolio that provide the highest returns, then invest in these. For example, the recipient could invest to increase trade activity with the USA, and in the related governance quality and infrastructure.

Step (4): The investment should lead to higher returns in the form of increased aid income, which is re-invested into the asset investment portfolio; e.g., increased trade activity with the USA should lead to further aid receipts donated by the USA, which then feeds back into the donor’s aid allocation model for the following years. The herding phenomenon then compounds the effect.

Treating aid determinants like assets in a portfolio implies the existence of an optimal mix of such variables which provides maximum return for minimal risk. There are in fact two main models that can be used for asset portfolio analysis: Modern Portfolio Theory (MPT) and the Capital Asset Pricing Model (CAPM). The CAPM model is more robust with fewer inputs; whereas the MPT model, though elegant, loses some practicality from the attempt to find asset returns, volatilities and correlations.

Unfortunately, the CAPM model’s principal purpose is for modelling and pricing equity market assets and their equivalents, where risk and returns are measured against some trade index such as the FTSE100. There is no equivalent transparently priced market for aid determinants and, therefore, the CAPM model cannot be used here. Instead, MPT is used to illustrate the concept using a simulation.

Let us assume that the portfolio for aid recipient $r_{j}$ contains two controllable assets, N=2, ‘owned’ by the recipient: trade activity, $t_{t}^{j},$ and poverty, $p_{t}^{j},$ at time subscript t, denoted in a set by
(11)$$P_{t}^{j}=\left\{t_{t}^{j}\right.,\left.p_{t}^{j}\right\}$$

These assets are ‘investable’ with varying risk-reward ratios and could be correlated or uncorrelated since increasing trade volumes do not always translate into reducing poverty, dependent on the recipient country and its regime as discussed earlier. For simplicity in this simulation, it is assumed that the assets are uncorrelated (ρ=0); however, equations can be adapted for the case when the assets are correlated and the correlation coefficient ρ≠0, discussed in Section 3.1.

The mean and variance of the two-asset portfolio (11) can be written as
(12)$$\mu_{P_{t}^{j}}=W\mu_{t^{j}}+(1-W)\mu_{p^{j}}$$
(13)$$\sigma_{P_{t}^{j}}^{2}=W^{2}\sigma_{t^{j}}^{2}+2W\left(1-W\right)\rho_{t_{t}^{j},p_{t}^{j}}\sigma_{t^{j}}\sigma_{p^{j}}+{\left(1-W\right)}^{2}\sigma_{p^{j}}^{2}$$
with the correlation between the assets subject to the constraint $-1{\leq \rho }_{t_{t}^{j},p_{t}^{j}}\leq 1$.

In (12) and (13), W∈ [0,1] is a parameter that determines the proportion of aid receipts invested in trade activity, i.e., W is the weight of the trade activity ‘asset’, $t_{t}^{j},$ in the portfolio. The weight on the poverty alleviation ‘asset’, $p_{t}^{j}$, must be 1−W, because
(14)$$\sum_{i=1}^{N}W_{i}=1$$

Further, if the two assets are uncorrelated, then $\rho_{t_{t}^{j},p_{t}^{j}}=0$ and the variance (13) becomes
(15)$$\sigma_{P_{t}^{j}}^{2}=W^{2}\sigma_{t^{j}}^{2}+{\left(1-W\right)}^{2}\sigma_{p^{j}}^{2}$$
The value of parameter W is important since the mean and standard deviation of returns of each asset should technically be known. As W is varied, the risk and reward dynamics of the portfolio change in response.

The minimum variance of the portfolio (‘risk-minimising portfolio’) is calculated by setting the first derivative of (15) to zero:(16)$$\frac{\partial \sigma_{P_{t}^{j}}^{2}}{\partial W}=2W\sigma_{t^{j}}^{2}+2\left(1-2W\right)\rho_{t_{t}^{j},p_{t}^{j}}\sigma_{t^{j}}\sigma_{p^{j}}-2\left(1-W\right)\sigma_{p^{j}}^{2}:=0$$
from which the value of W at the minimum can be found by (17):(17)$$W=\frac{\sigma_{p^{j}}^{2}-\rho_{t_{t}^{j},p_{t}^{j}}\sigma_{t^{j}}\sigma_{p^{j}}}{\sigma_{t^{j}}^{2}+\sigma_{p^{j}}^{2}-2\rho_{t_{t}^{j},p_{t}^{j}}\sigma_{t^{j}}\sigma_{p^{j}}}$$

For uncorrelated assets, (17) can be simplified to
(18)$$W=\frac{\sigma_{p^{j}}^{2}}{\sigma_{t^{j}}^{2}+\sigma_{p^{j}}^{2}}$$

## 3. Results

Let us start with an initial amount to invest; for simplicity, assume recipient $r_{j}$ invests 90% of total aid it receives with the remaining 10% lost to errors, corruption and aid spillage. Then, investment of these aid receipts in portfolio $P^{j}$ is defined by 0.9W in asset $t^{j}$ and 0.1(1−W) in asset $p^{j}$.

It is assumed that recipient $r_{j}$ is a rational investor and aims to maximise portfolio returns for the minimum risk (variance). It is further assumed that investment in its trade activity asset results in an increase in trade levels with donors, and an investment in its poverty alleviation asset at least maintains the current poverty level due to rising populations. Additionally, an investment in trade is assumed to be a risky investment, since money may be lost in the process, and poverty alleviation is deemed relatively risk-free since if it does not work, aid receipts should continue at the current level.

The values of the means, μ, and standard deviations, σ, for both portfolio assets to be input into the MPT Equations (12)–(18) can be calculated from the asset returns for each period, t. Returns on assets are usually calculated by taking the difference between the current asset value and the previous period’s asset value, the periodic asset income, and dividing by the previous period’s asset value, i.e., for the trade activity asset:(19)$$R_{t_{t}^{j}}=\frac{t_{t}^{j}-t_{t-1}^{j}}{t_{t-1}^{j}}$$
Such returns can be calculated for each time-period, t, starting at t=1. The mean and standard deviations of these returns can then be calculated.

However, using (19) is too simplistic for this simulation since trade volumes and poverty levels change substantially each period for many reasons, not solely due to any ‘return’ on an investment in these assets. The return arising purely from investment in these assets needs to be isolated from any additional ‘noise’ which may be causing their values to change. A recipient could isolate this ‘pure’ return using knowledge of the dynamics of their trade activity and poverty levels, controlling for the impact from any other variables. For the purposes of this simulation, it is assumed that this has been done by the chosen recipient, Ethiopia, resulting in the figures in Table 3.

In the MPT model Formulas (11)–(18), set recipient $r_{j}$ to be Ethiopia, j=E. It is also to be assumed that the portfolio asset returns in (11) are uncorrelated, $\rho_{t_{t}^{E},p_{t}^{E}}=0$, and therefore Equations (12), (15) and (18) are applicable to this simulation. Inputting the values from Table 3 into these equations creates a line in risk-return space, parameterised by W, which can be plotted, thus sketching out a hyperbola as W is varied. See Figure 3.

PQR of the concave function in Figure 3 is termed the efficient frontier. Choosing a portfolio mix that fits this line results in an optimum portfolio from a risk vs. reward perspective. Ethiopia could choose a portfolio mix anywhere in the risk-reward space in Figure 3. The efficient frontier identifies the possible portfolios that have the highest return for the least possible risk for that return in this risk-reward space.

In general terms, the efficient frontier contains portfolios which mathematically can be defined as solving
(20)$$\sigma_{P}=min\sqrt{\sum_{i=1}^{N}\sum_{j=1}^{N}W_{i}W_{j}\rho_{ij}\sigma_{i}\sigma_{j}}$$
subject to the constraints
(21)$$\mu_{P}=\sum_{i=1}^{N}W_{i}\mu_{i}$$
(22)$$\sum_{i=1}^{N}W_{i}=1$$

Ethiopia’s risk preferences will dictate where it wants to be on the curve in Figure 3. Clearly, Q is a better portfolio mix than O since they share similar levels of risk, for which Q offers the higher reward. A rational investor will always choose portfolio Q over O.

The values in Table 3 imply that investing in trade activity is higher risk, but provides a higher return, than poverty. If Ethiopia wanted to maximise return, it would choose portfolio R; or to minimise risk, portfolio P would be the best option. It depends on the recipient’s preferences and risk appetite.

If Ethiopia wanted to minimise risk, the minimum variance portfolio (MVP) should be targeted, which is calculated using Equation (18) as W=13.8%. This means that Ethiopia, with the risk and return characteristics in Table 3, should invest 13.8% of its 90% aid income into trade activities, with the remaining invested in poverty alleviation, producing an expected portfolio return of $\mu_{P_{t}^{j}}=1.14\%$ and a portfolio risk of $\sigma_{P_{t}^{j}}=0.19\%$,

Or, Ethiopia can choose portfolio R, maximizing both return and risk, for which it would invest all its available aid receipts into trade activity and nothing into poverty alleviation. There is no correct answer as to which portfolio mix Ethiopia should invest in, except that it should be one lying on the efficient frontier, PQR, in Figure 3.

For the purposes of simulating this model, assume Ethiopia decides to invest 50:50, so W=0.5. This means the portfolio chosen is one to the right of portfolio Q in Figure 3 on the efficient frontier. This portfolio has a return of 1.5%, with 1% originating from the investment in trade activity, $W{µ}_{t^{j}}=0.5\times 2$ and 0.5% from poverty reduction $(1-W{)µ}_{p^{j}}=0.5\times 1$, leading to an overall portfolio risk of 0.27%.

Next, the weighted network model (Equations (5)–(9)) is adapted to incorporate the described investment portfolio, with the year 2015 being t=1, and assuming the aid receipts for Ethiopia in 2014 were invested in trade and poverty assets for the year 2015 in accordance with the optimal portfolio mix. Then, using the inputs and data as described, including the values of the power parameters from Table 2, the model is iterated forward.

Throughout the simulation, the amount of aid in US$ donated in the years 2015 to 2019 by the three donors was fixed, as were the data fed into the functional input equations relating to Bangladesh and Afghanistan. Simply, the simulation adapts the model by using the MPT approach applied to Ethiopia only, which will result in different aid allocation percentages for all recipients for each period, compared to actual historical receipts. These updated allocations are fed back into the model at each annual iteration. By keeping all else fixed, the impact of an investment portfolio approach by Ethiopia, which will affect all recipients’ aid receipts, can be isolated.

Results of the simulation are presented in Figure 4, demonstrating the impact of Ethiopia investing its aid donations into trade activity and poverty alleviation in an optimum portfolio.

Figure 4 demonstrates that if Ethiopia had taken the MPT approach in 2015, as detailed above, then its aid receipts over the period 2015 to 2019 would be 48% higher than they were as a result of increasing trade activity further compounded by the bandwagon effect. This would come at the loss of aid receipts experienced by Bangladesh of 28% and Afghanistan of 24%.

Figure 5 shows the aid donated by each donor to each recipient in 2019 (t=5), including the total, demonstrating that Ethiopia is the ‘winner’ in terms of aid donations from all donors, although the UK allocations have not altered much because its parameters in Table 2 for 2019 indicate close to a uniform allocation per aid determinant, i.e., low biases for these aid determinants were shown by the UK in 2019 bordering on ambivalence.

Figure 6 below shows the total share of aid allocations from 2015 (t=1) to 2019 (t=5) using pie charts set side by side, with modelled aid donations (ODA) on the left and actual donations on the right. Ethiopia’s share would have increased from 34% actual to 50% as modelled, if it had invested in trade activity and poverty alleviation in 2015 using MPT, creating an optimum portfolio of these assets, further magnified by the herding phenomenon captured by the model.

This simulation has clearly demonstrated how the weighted network model can be adapted and incorporate MPT to be used as an investment tool by recipients to maximise their aid income.

### 3.1. Correlated Portfolio Assets

In the MPT simulation, it was assumed for simplicity of demonstration that the assets in Ethiopia’s portfolio were uncorrelated. However, what if they were in fact correlated? Assuming high governance quality and no corruption, increasing trade activity experienced by Ethiopia would be expected to reduce its poverty levels. Hence the returns from two assets, trade and poverty, assumed in the simulation should be negatively correlated.

Using MPT, the creation of an optimum portfolio follows the same process as detailed earlier, except now with a value for the correlation function, ρ, included in the Equations (12), (13) and (17). Assuming $\rho_{t^{E},p^{E}}=-0.8$, defining strong negative correlation between the trade and poverty assets in the portfolio, and using the variables as defined in Table 3, the same process performed in the simulation should be followed.

The presence of correlation produces a different efficient frontier curve than when there is no correlation, as shown in Figure 7 and compared to Figure 3. The shape of the curve has a large influence on the optimal portfolio.

The portfolio at point P, in Figure 7, is the minimal variance portfolio (MVP) for which the weight W is calculated using Equation (17), with values from Table 3 and $\rho_{t^{E},p^{E}}=-0.8$:(23)$$W=\frac{\sigma_{p^{E}}^{2}-\rho_{t^{E},p^{E}}\sigma_{t^{E}}\sigma_{p^{E}}}{\sigma_{t^{E}}^{2}+\sigma_{p^{E}}^{2}-2\rho_{t^{E},p^{E}}\sigma_{t^{E}}\sigma_{p^{E}}}=27\%$$
Using W=27%, the MVP has an expected return of 1.27% and portfolio risk of 0.09%.

The effect of correlation on portfolio volatility (which can be measured either by the standard deviation of portfolio returns, as here, or the variance) for this asset portfolio can be seen in Figure 8 showing a monotonic increasing function with an upper and lower bound of volatility when the asset correlation is 1 and −1, respectively. The function plotted is (24) using the inputs from Table 3, Equation (23), and $\rho_{t^{E},p^{E}}=-0.8$ (c.f. Equation (13)).
(24)$$\sigma_{P^{E}}=\sqrt{W^{2}\sigma_{t^{E}}^{2}+2W\left(1-W\right)\rho_{t^{E},p^{E}}\sigma_{t^{E}}\sigma_{p^{E}}+{\left(1-W\right)}^{2}\sigma_{p^{E}}^{2}}$$

## 4. Conclusions and Discussion

This paper explores the ability and effectiveness of a mathematical model, grounded in network theory, to capture the properties, dynamics and inter-dependencies inherent in foreign aid networks, and replicate a variety of donor and recipient behaviours. By doing so, this progresses the narrative around the analysis of foreign aid, illustrating how mathematics can be used to reveal useful features and intricate properties of real-world foreign aid networks.

Until now, regression analysis has been the dominant method used to analyse donor behaviour ([5,6,7]). However, here, data analysis of real-world foreign aid donations identifies the key parties involved and reveals their complex interactions. The network model developed can then also be used to investigate historical donor behaviour, akin to regression analysis, informing on the relative values of aid determinants used and their contribution to the donor’s final aid allocation decision. Nation donors do not publicise many of the determinants used in their allocation decisions; hence the data analysis described and performed was vital to identify the relevant variables and their interdependencies to provide input into this model. Furthermore, the parameter values $\mu_{\alpha }^{i}\;\mathrm{and}\;\eta_{\beta }^{i}$ contain information that reflect past motives and biases of a donor, and the relative importance the donor places on certain aid determinants.

The model was demonstrated to be flexible and adaptable enough to be used by aid recipients and donors. As an example of the model’s use, it was shown how Ethiopia could create a portfolio of assets based on this recipient’s determinants and apply modern portfolio theory to maximise aid income.

The approach can potentially be used by donors to replicate the properties of their foreign aid network and apply weights which control the allocation of their aid budget according to their own motivations and biases. Significantly, the weights that a donor would use is then explainable to the public, providing a transparent means of communication for politicians to justify their motivations behind their aid allocation decisions, and the model can also be iterated forward in time enabling a feedback mechanism to occur, in which donors (and recipients) can see the impact of their decisions on future aid allocations. Moreover, a donor could also treat the model like an investment tool, requiring a certain level of ‘return’ on their aid donations that can be quantified. To do so, donors would first create their specific foreign aid network model using the network science tools described earlier. The next step would be to adapt the model Equations (1)–(3) to include chosen determinants and their parameter values which reflect their motivations and the investments they want to make. An obvious one is trade activity: donors can use the model to ensure that more aid is allocated to countries that provide higher levels of trade activity with the donor, such as that seen between Germany and China [17].

Finally, by providing a framework to explore the properties of foreign aid networks and the impact that decision variables will have on those properties, including the final aid allocations, the weighted network model can help donors and recipients, and potentially multilateral organisations, with one of the issues associated with foreign aid: aid spillage, by reducing the costs arising from inappropriate use of foreign aid budgets.

In conclusion, the weighted network model, underpinned by network theory, has been demonstrated to successfully model the international aid system and is able to shed new light on the complexity and interactions inherent in foreign aid networks.

## Figures and Tables

**Figure 1 entropy-25-00641-f001:**
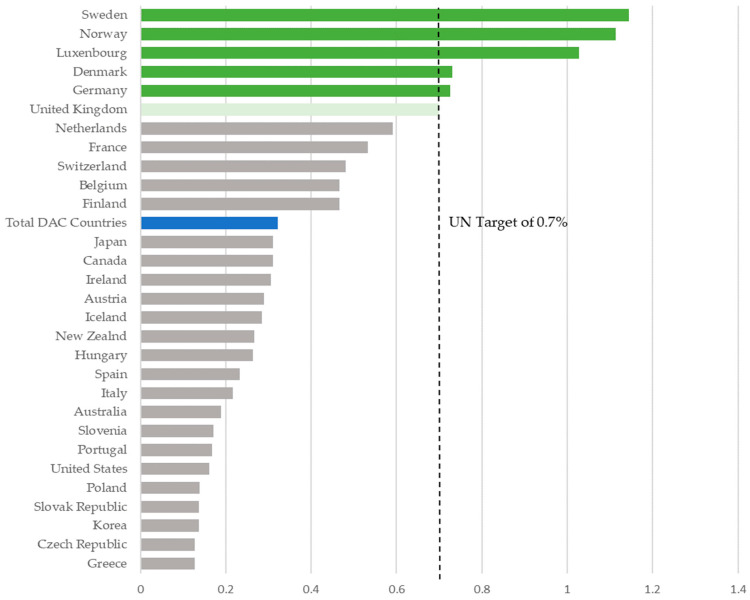
ODA grant equivalent as percentage of GNI in 2020 for DAC donors in the OECD. The grey bars identify those countries that contribute less than the United Nations (UN) target of 0.7%, the blue bar shows total of the DAC countries as a percentage of their GNI and the green bars highlight those countries who contribute over the UN target of 0.7%.

**Figure 2 entropy-25-00641-f002:**
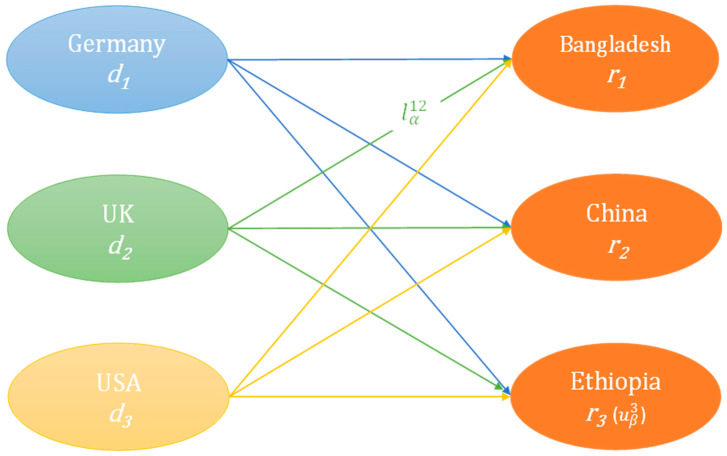
A complete bipartite graph of a weighted network containing notation which underpins the model.

**Figure 3 entropy-25-00641-f003:**
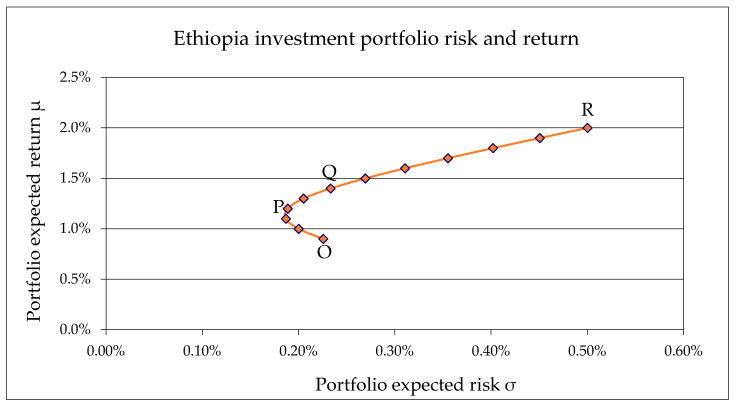
Ethiopia’s investment portfolio risk and return profile for two uncorrelated assets. A hyperbola is created from varying the relative weights of the portfolio assets resulting in different risk and reward profiles. The top of the hairpin, PQR, is the efficient market frontier containing optimised portfolio asset weights. The point O represents the portfolio with the lowest return.

**Figure 4 entropy-25-00641-f004:**
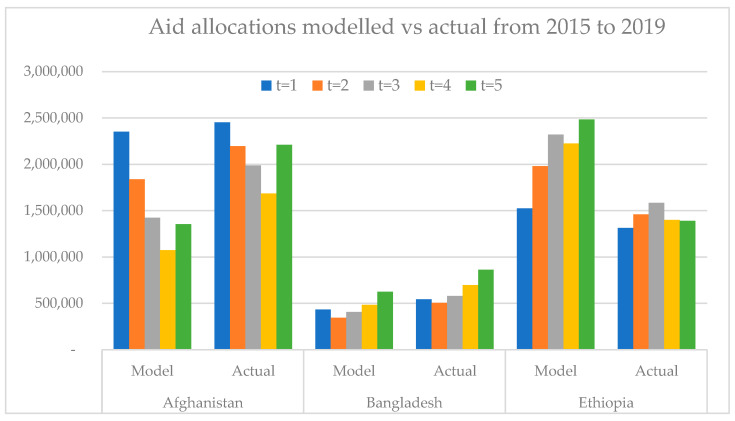
Aid allocations from 2015 (t=1) to 2019 (t=5), modelled by simulating the adapted model using Equations (5)–(9), adjusted for MPT, and comparing to actual aid donated. The model results incorporate Ethiopia’s aid investments in an optimum portfolio.

**Figure 5 entropy-25-00641-f005:**
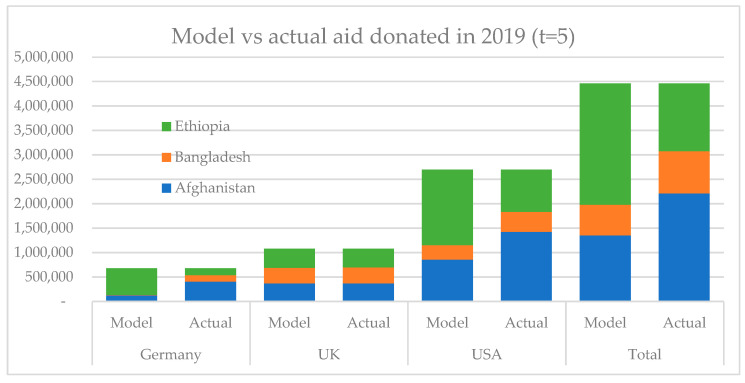
Aid allocations modelled vs. actual in 2019 (t=5) by donor to each recipient.

**Figure 6 entropy-25-00641-f006:**
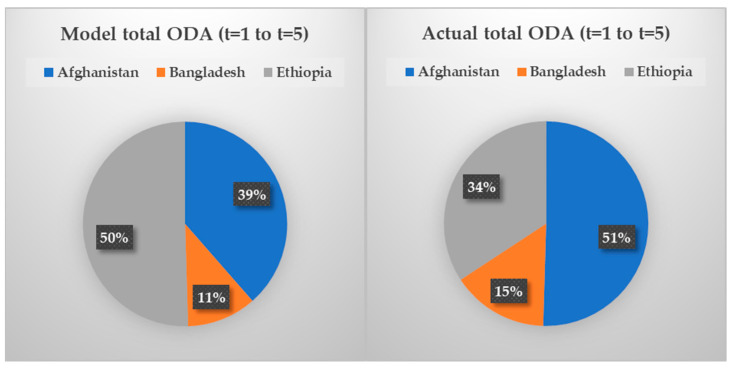
Pie charts of the total aid allocation percentages modelled and actual for the period from 2015 (t=1) to 2019 (t=5) per aid recipient.

**Figure 7 entropy-25-00641-f007:**
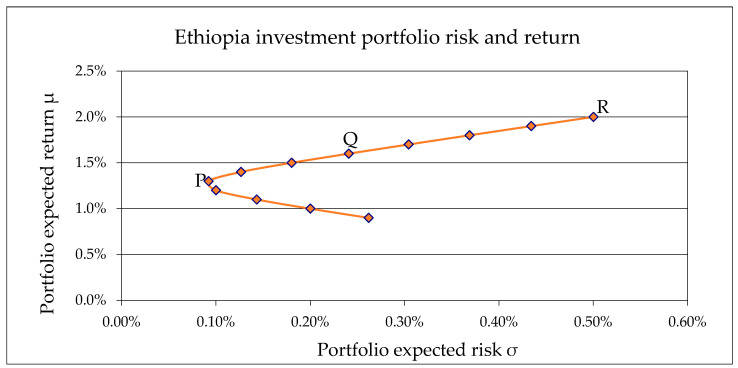
Ethiopia investment portfolio risk and return. Compared to Figure 3, the presence of asset correlation produces a different risk and reward profile, providing different optimum portfolio combinations.

**Figure 8 entropy-25-00641-f008:**
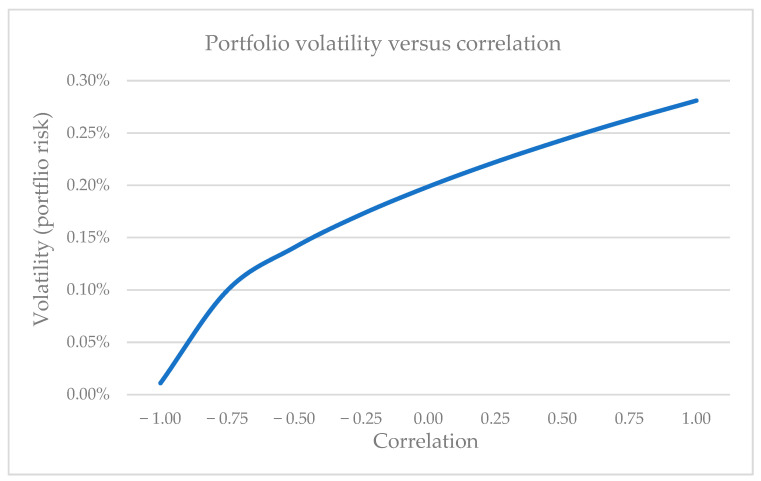
Volatility of the portfolio against correlation of the assets in the portfolio.

**Table 1 entropy-25-00641-t001:** Top 10 ODA recipients, including significant regional aid donations, and figures for all developing counties for comparison [9]. All figures in US$m unless otherwise stated.

	Net ODA Receipts					GNI/CAP (US$)	GNI	ODA/GNI (%)
Country/Region	2015	2016	2017	2018	2019	2019	2019	2019
Syrian Arab Republic	4920	8900	10,428	9997	10,252	-	-	-
Ethiopia	3239	4084	4125	4941	4810	850	95,641	5.03
Bangladesh	2593	2533	3782	3045	4518	1940	316,907	1.43
Yemen	1778	2301	3234	7985	4397	-	-	-
Afghanistan	4274	4069	3812	3792	4285	540	19,402	22.08
Nigeria	2432	2498	3359	3305	3531	2030	433,449	0.81
Kenya	2464	2188	2480	2491	3251	1750	93,578	3.47
Democratic Republic of the Congo	2599	2102	2293	2514	3026	520	45,879	6.59
Jordan	2141	2728	2980	2526	2797	4300	43,429	6.44
India	3174	2679	3198	2462	2611	2130	2,843,902	0.09
**Regional (not specific to any country)**								
South of Sahara	2435	2635	2759	3137	3410			
Africa region	2184	2777	3017	3241	3201			
**All developing countries**	146,742	158,811	165,090	166,540	168,588	511,750	292,854,611	0.58

**Table 2 entropy-25-00641-t002:** Power parameters required by the model to recreate the actual aid allocation results for each year 2015 to 2019 for donors Germany, UK and USA and recipients Afghanistan, Bangladesh and Ethiopia. Data that is not relevant is labelled N/A for not applicable.

Donor	Aid Determinant	2015	2016	2017	2018	2019
Germany	Colonial history	N/A	N/A	N/A	N/A	N/A
	Trade relationship	1.5	1.4	2.0	0.8	-
	Poverty	1.1	1.6	1.1	2.0	1.0
	Bandwagon	4.0	6.0	9.0	5.0	2.2
UK	Colonial history	-	-	0.5	-	-
	Trade relationship	0.4	0.4	0.4	0.1	-
	Poverty	-	-	0.6	0.2	0.1
	Bandwagon	1.0	1.0	0.6	0.3	0.1
USA	Colonial history	N/A	N/A	N/A	N/A	N/A
	Trade relationship	-	-	-	1.1	-
	Poverty	1.4	1.6	1.5	1.6	0.5
	Bandwagon	1.0	0.6	0.4	0.3	0.9

**Table 3 entropy-25-00641-t003:** Parameters relating to Ethiopia’s investment in its portfolio of assets.

$\mu_{t^{j}}$: Return on Trade	$\sigma_{t^{j}}$: Risk of Trade	$\mu_{p^{j}}$: Return on Poverty	$\sigma_{p^{j}}$: Risk on Poverty
2.0	0.5	1.0	0.2

## Data Availability

Data regarding aid donations are obtained from reference [1].

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
