# Peer review of "A Network Model Approach to International Aid"

_entropy, 2023, doi:10.3390/e25040641_

Round 1

Reviewer 1 Report

Excellent paper, very well written, novel methodology, well presented results. 

Should be published largely as is.

Notes: Fig 1 mentions color - do authors intend to pay for color printing? Similarly for fig 2 - why is it shaded?

2.1.2 onwards - Authors could mention that many others factors could be included in model - like corruption or war. Instead they imply that their factors are the only ones.

Table 2 is irrelevant.

Unfortunately the chosen data includes numerous missing variables - as shown in Table 3. This does raise serious issues and should be mentioned as  a limitation in their example, eg; an attempt to create a trend from US trade relationships from 1 variable are dubious. Are better examples available?

Author Response

Response to Reviewer 1

Many thanks for sparing the time to look at our research article. We were pleased to hear that you thought that the paper had merit. The manuscript has been revised in response to your comments.

We are grateful for your comments aimed at improving the presentation of the results, and we offer the follow comments in response to your review.

1) Regarding the use of colour in Figure 1: the journal has informed me that there is no charge for the use of colour and, since the colour helps us to differentiate between the various categories. its use is retained in the revised manuscript.

2) Regarding the factors included in the model: We agree, and we are grateful that you encourage us to make this clearer in the text. A new paragraph in Section 2.2 from lines 182-188 has now been added making it clear that other factors could play a part.

3) Regarding Table 2: We agree and apologise. This information had been gathered early in the research but is not needed here. The table has been removed and tables re-numbered accordingly.

4) Regarding the missing data in Table 3: This is now Table 2 following the comment above. The dashed line in the previous version makes it look as if the data was missing but in fact it would be better to say that the data was not applicable and so this has been changed in the table and a comment made in line 372-373.

Reviewer 2 Report

In this paper authors develop a weighted network approach to the international aid. Their method is shown to offer some advantages over more traditional techniques in this field based on regression or least-square methods. The paper is well written, results are moderately interesting and it might be published in the present form.

Author Response

Response to Reviewer 2

Many thanks for sparing the time to look at our research article. The manuscript has been revised in response to comments from other reviewers. Although there is still much work that can be done, we were pleased to hear that you thought that the paper had merit.